# Recovering the Free-Field Acoustic Characteristics of a Vibrating Structure from Bounded Noisy Underwater Environments

**DOI:** 10.3390/s21165521

**Published:** 2021-08-17

**Authors:** Wei Lin, Sheng Li

**Affiliations:** 1Qingdao Innovation and Development Center, Harbin Engineering University, Qingdao 266000, China; linwei23@hrbeu.edu.cn; 2State Key Laboratory of Structural Analysis for Industrial Equipment, School of Naval Architecture, Dalian University of Technology, Dalian 116024, China

**Keywords:** free field recovery, free-field acoustic characteristics, bounded noisy underwater environment, boundary element method, vibro-acoustic coupling method

## Abstract

The vibrational behavior of an underwater structure in the free field is different from that in bounded noisy environments because the fluid–structure interaction is strong in the water and the vibration of the structure caused by disturbing fields (the reflections by boundaries and the fields radiated by sources of disturbances) cannot be ignored. The conventional free field recovery (FFR) technique can only be used to eliminate disturbing fields without considering the difference in the vibrational behavior of the structure in the free field and the complex environment. To recover the free-field acoustic characteristics of a structure from bounded noisy underwater environments, a method combining the boundary element method (BEM) with the vibro-acoustic coupling method is presented. First, the pressures on the measurement surface are obtained. Second, the outgoing sound field and the rigid body scattered sound field are calculated by BEM. Then, the vibro-acoustic coupling method is employed to calculate the elastically radiated scattered sound field. Finally, the sound field radiated by the structure in the free field is recovered by subtracting the rigid body scattered sound field and the elastically radiated scattered sound field from the outgoing sound field. The effectiveness of the proposed method is validated by simulation results.

## 1. Introduction

It is important to characterize a target in an intrinsic way for target identification and control [1]. To acquire the intrinsic acoustic characteristics of the target, the measurement must be carried out in the free field. However, there is no ideal free field in nature or laboratory. Although the sound field radiated by a target in a large lake or anechoic tank can be approximated as a free sound field, the measurement is susceptible to bio-acoustic background noise and climate change in the large lake [2], and it is difficult to satisfy the free-field condition in the anechoic tank at low frequencies. To recover the free-field acoustic characteristics in the non-anechoic environment, the FFR techniques have been developed.

Pachner [3] initially showed a method separating the traveling and standing components on the surfaces of two spheres surrounding the source. The outgoing and incoming components of the sound field were separated based on spherical harmonics for two concentric spheres when the sound source is located inside the inner sphere [4]. Tsukernikov [5] combined the Helmholtz integral equation and spherical wave expansion method to realize the sound field calculation in a closed space. Williams [6] showed an approach to remove the disturbing fields based on spatial Fourier transform. This method has been widely applied to nearfield acoustic holography (NAH) [7,8]. However, this method can only be used for regular measurement surfaces and sound sources such as planes, cylinders, and spheres. The statistically optimized nearfield acoustic holography (SONAH) was developed to predict the pressure and velocity in complex environments [9,10]. However, this technique cannot accurately reconstruct the local sound field. To identify source velocities in complex environments, the inverse patch transfer functions method (iPTF) suitable for three-dimensional structures was proposed [11,12,13,14]. However, the problem of inversion of the ill-conditioned matrix needs to be solved when this method is used. The effects of both scattering (due to incident excitation) and radiation (due to interior noise sources) were removed throughout a “virtual” volumetric sonar array projected within the structure [15]. However, the capability to determine the range of near-field sources or scatterers needs to be improved. The supersonic intensity in a reverberant environment (SIRE) technique, which made use of an underwater vector sensor to obtain narrow-band sound power and directivity of a target in a reverberation environment was proposed [2]. However, the scattered sound field was not removed in this method. Langrenne et al. [16] presented a method based on BEM to recover the free sound field of a structure in a complex environment by considering the scattering effects on the machine. The equivalent source method (ESM)-based FFR technique combined with the NAH was used to reconstruct the sound field radiated by a target source from the measured mixed field [17,18,19,20]. An improved double-layer BEM was proposed to recover the half-space acoustic characteristics of a target source sitting on the boundaries in a bounded noisy environment by combining it with the image-source method [21]. Wu et al. [22] presented a BEM-based NAH in conjunction with the FFR technique to reconstruct the free sound field in a non-anechoic environment. However, the interaction between the structure and the fluid medium is not considered in these four methods. Although Sternini et al. [23] presented a method taking into account the vibrational response of an elastic object caused by incident wave, this method was used to calculate the bistatic scattered and needed to solve the problem of singular matrix inversion.

The conventional FFR technique does not take care of the difference between the vibrational behaviors of the structure in the free field and in the bounded noisy underwater environment. A method combining the BEM with the vibro-acoustic coupling method is proposed to recover the free-field acoustic characteristics in a bounded noisy underwater environment by considering the fluid–structure interaction. The goal of this article is to recover the free-field acoustic characteristics of the vibrating structure rather than to reconstruct the surface vibration of the structure. Thus, the NAH is not required. The theory of the proposed method is described in detail in Section 2. Section 3 validates the effectiveness of the proposed method through numerical simulations. Conclusions and suggestions for future research efforts are given in Section 4.

## 2. Theory

Figure 1 shows a structure vibrating in the free field and in a bounded noisy environment. As shown in Figure 1, the vibration of the structure is only caused by the force in the free field. However, the structural vibration in the bounded noisy environment is caused not only by the force but also by the disturbing fields (the reflections by boundaries and the fields radiated by sources of disturbances) [24]. For clarity, the vibrational behavior of the structure in the free field is defined as $V_{1}$ and the vibrational behavior of the structure in the bounded noisy environment is defined as $V_{2}$ in this paper. The elastic vibration of the structure caused by disturbing fields might be ignored in the air because the interaction between the structure and the air is weak. In this case, $V_{1}$ is the same as $V_{2}$. However, the elastic vibration of the structure caused by disturbing fields is not ignorable in the water due to the strong interaction between the structure and the water. This means that $V_{1}$ is different from $V_{2}$ in the water, and the sound field radiated by the structure in the free field is different from that in the bounded noisy environment. Thus, to recover the free-field acoustic characteristics radiated by the structure in the bounded noisy underwater environment, the vibration of the structure caused by disturbing fields must be considered, especially when the structure is a kind of thin-shell structure.

### 2.1. Sound Field Separation

For a vibrating structure in a complex underwater environment, as shown in Figure 2, $\Gamma_{1}$ is the surface of structure $S_{0}$; $\Gamma_{2}$ represents a boundary of the sound field; *S* is the measurement surface; the space between $\Gamma_{1}$ and *S* is $A_{1}$, and the other space is $A_{2}$; Q(r) represents the source of disturbance; and $n_{\Gamma_{1}}$, $n_{\Gamma_{2}}$, and $n_{S}$ are the normals to $\Gamma_{1}$, $\Gamma_{2}$, and *S*, respectively.

The sound pressure p(r) in a sound field is composed of incoming sound pressure $p^{i}\left(r\right)$ and outgoing sound pressure $p^{o}\left(r\right)$. It can be written as follows [16]:(1)$$p\left(r\right)=p^{i}\left(r\right)+p^{o}\left(r\right)=p^{i}\left(r\right)+p^{f}\left(r\right)+p^{s}\left(r\right),$$

It should be noted that $p^{i}\left(r\right)$ is composed of the sound pressure radiated by all sources of disturbances and reflected by the boundaries and that $p^{o}\left(r\right)$ consists of the free sound pressure $p^{f}\left(r\right)$ and the scattered sound pressure $p^{s}\left(r\right)$.

The scattered field is the superposition of an elastic contribution and a rigid contribution from the structure when sound waves are incident on the structure [23]. Thus, the scattered sound pressure $p^{s}\left(r\right)$ can be written as
(2)$$p^{s}\left(r\right)=p^{sr}\left(r\right)+p^{se}\left(r\right),$$
where $p^{sr}$ is the pressure scattered from an infinitely rigid body (rigid body scattered sound pressure) and $p^{se}$ is the scattered component due to the elasticity of the structure (elastically radiated sound pressure) [23]. In the double-layer BEM [16], $p^{se}$ can be ignored due to industrial sources have much lower admittances than air. However, $p^{se}$ is not ignorable in the water because of the strong interaction between the structure and the water.

When the field points in the $A_{1}$, this is an interior problem. The outgoing sound pressure $p^{o}\left(r\right)$ can be calculated from [16]
(3)$$\begin{array}{cc} \; & p^{o}\left(r\right)-\int_{S}\left[p\left(s^{\prime }\right)\partial_{n}G\left(r,s^{\prime }\right)+i\rho \omega v_{n}\left(s^{\prime }\right)G\left(r,s^{\prime }\right)\right]dS\\ \; & =\left\{\begin{array}{c} p\left(r\right)\;\;\;\;for\;r\in A_{1}\\ c(r)p(r)\;\;for\;r\in S\\ 0\;\;\;for\;r\in A_{2}, \end{array}\right. \end{array}$$
where *i*, ρ, ω, and $G(r,s^{\prime })$ are the imaginary unit, the fluid density, the circular frequency, and the three-dimensional free-space Green’s function for Helmholtz equation, respectively. c(r) is solid angle coefficient and is given by [25]
(4)$$c\left(r\right)=-\int_{S}\frac{\partial }{\partial n}\left(\frac{1}{4\pi r}\right)dS,$$

The sound pressure $p(s^{\prime })$ and normal velocity $v_{n}\left(s^{\prime }\right)$ on the measurement surface *S* can be calculated from the sound pressures on two parallel closed surfaces surrounding the structure [16]. Then, the $p^{o}\left(s^{\prime }\right)$ on the *S* can be obtained from the Equation (Equation 3).

When the $p^{o}\left(s^{\prime }\right)$ is known, only $p^{s}\left(s^{\prime }\right)$ needs to be calculated to obtain $p^{f}\left(s^{\prime }\right)$. To calculate the scattered field, the incident field on the $\Gamma_{1}$ is necessary. When the field points in the $A_{2}$, this is an exterior problem. For the exterior problem, the incoming sound pressure on the $\Gamma_{1}$ can be solved by [16]
(5)$$\begin{array}{cc} \; & p^{i}\left(r\right)+\int_{S}\left[p\left(s^{\prime }\right)\partial_{n}G\left(r,s^{\prime }\right)+i\rho \omega v_{n}\left(s^{\prime }\right)G\left(r,s^{\prime }\right)\right]dS\\ \; & =\left\{\begin{array}{c} 0\;\;\;\;\;\;for\;r\in A_{1}\\ (1-C^{0}\left(p\right))p\left(r\right)\;\;for\;r\in S\\ p\left(r\right)\;\;\;\;for\;r\in A_{2}. \end{array}\right. \end{array}$$

### 2.2. Subtraction of the Scattered Field

The Helmholtz integral equation for a scattered problem can be expressed as [25]
(6)$$\left(1-c^{\prime }\left(s\right)\right)p^{\prime }\left(s\right)=p^{i}\left(s\right)-\int_{\Gamma_{1}}\left[p^{\prime }\left(s\right)\partial_{n}G\left(r,s\right)+i\rho \omega v_{n}^{\prime }\left(s\right)G\left(r,s\right)\right]dS,\;r\in A_{1},A_{2}\;,$$
where $p^{\prime }\left(s\right)$ and $v_{n}^{\prime }\left(s\right)$ are respectively correspondent with the sound pressure and normal velocity on the $\Gamma_{1}$ at *s* and where $c^{\prime }\left(s\right)$ is coefficient on the $\Gamma_{1}$.

According to the definition of the rigid body scattered sound pressure, $v_{n}$ equals zero. Then, the Equation (Equation 6) can be simplified as [16]
(7)$$\left(1-c^{\prime }\left(s\right)\right)p^{b}\left(s\right)=p^{i}\left(s\right)-\int_{\Gamma_{1}}\left[p^{b}\left(s\right)\partial_{n}G\left(r,s\right)\right]dS,\;\;r\in \Gamma_{1},$$
where $p^{b}$ is the blocked pressure, which is the sum of $p^{sr}$ and $p^{i}$[23]. When $p^{i}$ is known, $p^{b}$ can be calculated by Equation (Equation 7). Furthermore, $p^{sr}$ on the *S* radiated from $\Gamma_{1}$ can be computed by
(8)$$p^{sr}\left(r\right)=-\int_{\Gamma_{1}}\left[p^{b}\left(s\right)\partial_{n}G\left(r,s\right)\right]dS,\;\;r\in S,$$

The remaining problem is to calculate $p^{se}$ since $p^{sr}$ is obtained. As mentioned before, $p^{se}$ is radiated from elastic vibration of the structure caused by the incoming field. It is found that the elastic term $p^{se}$ is the elastically radiated component of the pressure field related to the elastic surface velocity $v_{n}^{se}$, and together, they represent a radiation problem only, with no incident field involved [23]. In other words, $p^{se}$ and $v_{n}^{se}$ can be expressed as
(9)$$\left(1-c^{\prime }\left(s\right)\right)p^{se}\left(s\right)=-\int_{\Gamma_{1}}\left[p^{se}\left(s\right)\partial_{n}G\left(r,s\right)+i\rho \omega v_{n}^{se}\left(s\right)G\left(r,s\right)\right]dS.$$

When $v_{n}^{se}$ is given, $p^{se}$ can be acquired. Then, the problem is converted to calculate $v_{n}^{se}$. In order to compute $v_{n}^{se}$, the fluid–structure coupling problem must be solved. The BEM has advantages in calculating the far-field sound pressure and can be used for fluid modeling [26]. In addition, the BEM can be coupled with the finite element method (FEM) to solve the fluid–structure coupling problem [27]. The coupled method is the vibro-acoustic coupling method. Considering that the indirect boundary element method (IBEM) can be used to calculate the internal and external sound fields at the same time, IBEM is coupled with FEM. First, the structure is discretized by FEM, and the corresponding finite element equation is [27]
(10)$$\left[\begin{array}{c} K_{s}+i\omega C_{s}-\omega^{2}M_{s} \end{array}\right]\left[\begin{array}{c} u \end{array}\right]=\left[\begin{array}{c} F_{s} \end{array}\right],$$
where *u* is the displacement of structural node; $M_{s}$, $C_{s}$, and $K_{s}$ are the mass matrix, the damping matrix, and the stiffness matrix of the structure, respectively; $F_{s}$ is the mechanical load acting on the structure; and the time component is $e^{i\omega t}$, where *t* is the time.

Then, the displacement and the sound field of the structure are solved simultaneously considering the continuity of the velocity at the coupling boundary. Ignoring the damping effect and satisfying the single layer potential, the Equation (Equation 10) coupled with IBEM can be expressed as [27]
(11)$$\left[\begin{array}{cc} K_{s}+i\omega C_{s}-\omega^{2}M_{s} & L_{C}\\ L_{C}^{T} & \frac{D(\omega )}{\rho \omega^{2}} \end{array}\right]\left[\begin{array}{c} u\\ \mu \end{array}\right]=\left[\begin{array}{c} F_{s}\\ F_{a} \end{array}\right],$$
where μ is the double-layer potential, that is, the sound pressure difference on the structure surface; $L_{C}$ is the vibro-acoustic coupling matrix; the superscript letter *T* stands for the transpose of matrix; D(ω) is the IBEM influence matrix; and $F_{a}$ is the acoustic load.

The mechanical load is equal zero when solving the scattered problem of the incident field on the elastic body. Additionally, it is noted that the acoustic load is approximated as $p^{b}\left(r\right)$ rather than $p^{i}\left(r\right)$. Equation (Equation 11) can be solved when the properties of structure are known before the sound field is recovered. Then, the surface velocity $v_{n}^{se}$ can be obtained from *u*. The $p^{se}\left(r\right)$ on the *S* can be calculated by the Helmholtz integral equation:(12)$$\begin{array}{cc} \; & -\int_{\Gamma_{1}}\left[p^{se}\left(s\right)\partial_{n}G\left(r,s\right)+i\rho \omega v_{n}^{se}\left(s\right)G\left(r,s\right)\right]dS\\ \; & =\left\{\begin{array}{c} p^{se}\left(r\right)\;\;\;\;,\;r\in A_{1},A_{2}\\ \left(1-c^{\prime }\left(s\right)\right)p^{se}\left(s\right)\;\;,\;s\in \Gamma_{1} \end{array}\right. \end{array}.$$

Using pressure fields $p^{o}\left(r\right)$, $p^{sr}\left(r\right)$, and $p^{se}\left(r\right)$ computed from Equations (Equation 3), (Equation 8), and (Equation 12), respectively, the $p^{f}\left(r\right)$ is calculated from Equation (Equation 1):(13)$$p^{f}\left(r\right)=p^{o}\left(r\right)-p^{sr}\left(r\right)-p^{se}\left(r\right)=p_{V_{2}}^{f}\left(r\right)-p^{se}\left(r\right),\;r\in S$$
where $p_{V_{2}}^{f}\left(r\right)$ is the sound field radiated by the structure for which the vibrational behavior is $V_{2}$, and it is clear that $p_{V_{2}}^{f}\left(r\right)$ is the sound field recovered by the double-layer BEM.

### 2.3. Discretization

To calculate the integral equations, the sufaces are discretized into elements. Then, the outgoing and incoming sound fields can be rewritten as
(14)$$P_{S}^{o}=\left[C_{S}-H_{S}^{S}\right]P_{S}-i\rho \omega \left[G_{S}^{S}\right]V_{S},$$
(15)$$P_{\Gamma_{1}}^{i}=-\left[H_{S}^{\Gamma_{1}}\right]P_{S}-i\rho \omega \left[G_{S}^{\Gamma_{1}}\right]V_{S},$$
where *C*, *H*, and *G* are the solid angle coefficient matrix, the sound pressure, and the particle velocity transfer matrices, respectively. The subscript letter and the superscript letter stands for the integration surface and the target surface, respectively. Subsequently, the $P^{b}$ on the $\Gamma_{1}$ can be obtained from Equation (Equation 7):(16)$$P_{\Gamma_{1}}^{b}={\left[I-C_{\Gamma_{1}}^{\prime }+H_{\Gamma_{1}}^{\Gamma_{1}}\right]}^{-1}P_{\Gamma_{1}}^{i},$$
where *I* is the identity matrix and $C^{\prime }$ is the coefficient matrix.

The combined Helmholtz integral equation formulation (CHIEF) method is used to overcome the non-uniqueness difficulty [28,29]. According to Equation (Equation 8), the sound pressure scattered from an infinitely rigid body on the *S* can be given by
(17)$$P_{S}^{sg}=-\left[H_{\Gamma_{1}}^{S}\right]P_{\Gamma_{1}}^{b}.$$

In order to calculate the response of the structure caused by the incoming field, the mechanical load is zero and the acoustic load is $p^{b}$. The properties including mass, the damping, and the stiffness of structure have to be known in advance. Then, the Equation (Equation 11) can be rewritten as
(18)$$\left[\begin{array}{cc} K_{s}+i\omega C_{s}-\omega^{2}M_{s} & L_{C}\\ L_{C}^{T} & \frac{D(\omega )}{\rho \omega^{2}} \end{array}\right]\left[\begin{array}{c} U\\ \Phi \end{array}\right]=\left[\begin{array}{c} 0\\ P_{\Gamma_{1}}^{b} \end{array}\right].$$
where *U* is the displacement vector and Φ is the vector of the double-layer potential.

When the Equation (Equation 18) is solved, the normal velocity of structural surface $V_{\Gamma_{1}}^{se}$ can be achieved by
(19)$$V_{\Gamma_{1}}^{se}=\left[G\right]\left[U\right],$$
where *G* is the transformation matrix, the function of which is to convert the displacement vector *U* of the structural node into the normal velocity of the structure surface $V_{\Gamma_{1}}^{se}$. Furthermore, the $P_{\Gamma_{1}}^{se}$ can be achieved by
(20)$$P_{\Gamma_{1}}^{se}=-i\rho \omega {\left[I-C_{\Gamma_{1}}^{\prime }+H_{\Gamma_{1}}^{\Gamma_{1}}\right]}^{-1}\left[G_{\Gamma_{1}}^{\Gamma_{1}}\right]V_{\Gamma_{1}}^{se},$$

Then, the $P_{S}^{se}$ on the *S* can be calculated from Equation (Equation 12)
(21)$$P_{S}^{se}=-\left[H_{\Gamma_{1}}^{S}\right]P_{\Gamma_{1}}^{se}-i\rho \omega \left[G_{\Gamma_{1}}^{S}\right]V_{\Gamma_{1}}^{se},$$

Finally, the free sound field recovered by the proposed method in the bounded noisy underwater environment is obtained using
(22)$$P_{S}^{f}=P_{S}^{o}-P_{S}^{sr}-P_{S}^{se}.$$

## 3. Numerical Simulations

### 3.1. In a Bounded Noisy Air Environment

The numerical model is shown in Figure 3. The structure is a 1×1×1 m${}^{3}$ cubic shell and the side length a=1. The properties of the shell are shown in Table 1. The coordinate origin is at the centre of the shell. The structure is excited by a point force with amplitude 10 N and located at (0 m, 0 m, −0.5 m), which is the centre of the upper surface. The upper surface of the structure is simply supported. The fluid medium is air. The sound speed in the air and the fluid density are 340 m/s and 1.225 kg/m${}^{3}$, respectively. In the air, the reference sound pressure is $2\times 10^{-5}$ Pa, and the reference sound power is $1\times 10^{-12}$ W. There is an infinite rigid plane at z=1 m to simulate the rigid boundary such as the ground. A monopole located at (0.75 m, 0.75 m, 0.75 m) is used as a source of disturbance to influence the sound field. At 100 Hz, the strength of the monopole is −0.016i m${}^{3}$/s.

As the structure is a cubic shell, the strength of the fluid–structure coupling can be judged by the characteristic quantity λ [30]:(23)$$\lambda =\frac{\rho c}{\rho_{s}h\omega },$$
where *c*, *h*, and $\rho_{s}$ are the sound speed in the fluid, the thickness of the plate, and the density of the plate, respectively. When λ<1, the coupling between the structure and the fluid is very weak; when λ>1, the coupling between the structure and the fluid is strong. When the cubic shell vibrates in the air, λ is less than 0.1 in the studied frequency range. Thus, the coupling between the cubic shell and the air is very weak.

The dimensions of the measurement surface *S* are 1.2×1.2×1.2 m${}^{3}$. The distance between the two simulated layers located on each side of *S* is 0.06 m, which satisfies the convergence condition [20]. The measurement surface is discretized by 3456 linear quadrilateral elements with 3458 nodes. The structure is discretized by 2400 linear quadrilateral shell elements with 2402 nodes. The maximum frequency allowed by the mesh is 1133 Hz, which corresponds to ka=21 (*k* is the wavenumber) based on the well-known 1/6 criterion in the air. The maximum frequency allowed by the mesh is 5000 Hz, which corresponds to ka=21 based on the 1/6 criterion in underwater environments. It should be noted that, in the following numerical simulation, the sound pressure on the measurement surface is obtained from computation, not from real experimental testing.

First, the sound pressures on the measurement surface are calculated by the vibro-acoustic coupling method in the free field when the structure is excited by the point force. The sound field in this case is defined as the $V_{1}$ free sound field. Second, the sound pressures on the two simulated layers are calculated by the vibro-acoustic coupling method when the structure is excited by the force in the bounded noisy environment. In this case, the directly obtained sound field is defined as the total sound field. Furthermore, the normal velocities on the surface of the structure that vibrate in the bounded noisy environment are taken as a boundary condition to compute sound field in the free field. Then, the calculated sound field is defined as the $V_{2}$ free sound field. Finally, the sound field recovered by the double-layer BEM and the sound field recovered by the proposed method are defined as the conventional recovered sound field labeled as c_recovered sound field and the new recovered sound field labeled as n_recovered sound field, respectively.

The power estimator is defined as [16]:(24)$$I_{e}=\int_{S}\frac{|p^{e}{\left(s\right)|}^{2}}{\rho c}dS$$
where $p^{e}\left(s\right)$ is sound pressure in the sound field. When calculating the power estimator in $V_{1}$ free sound field, $p^{e}\left(s\right)$ is equal to $p^{f}\left(s\right)$.

The mean quadratic error is used to compare the differences of sound fields. It is defined as [16]
(25)$$E=\sqrt{\frac{\int_{S}{\left|p^{e}\left(s\right)-p^{F}\left(s\right)\right|}^{2}dS}{\int_{S}{\left|p^{F}\left(s\right)\right|}^{2}dS}}$$
where $p^{F}\left(s\right)$ is equal to $p^{f}\left(s\right)$, when calculating the mean quadratic error of sound field to $V_{1}$ free sound field. $p^{F}\left(s\right)$ is equal to $p_{V_{2}}^{f}$ when calculating the mean quadratic error of sound field to $V_{2}$ free sound field.

The power estimator levels directly obtained and recovered are shown in Figure 4. The power estimator level in the total sound field is very different from the power estimator level in the $V_{1}$ free sound field. The differences are more than 7 dB at most frequencies. This means that boundaries and sources of disturbances have a great influence on the acquisition of the free-field acoustic characteristics of the vibrating structure. However, the power estimator levels in the $V_{1}$ free sound field, in the $V_{2}$ free sound field, and in the conventional recovered sound field are the same, and the differences are less than 1 dB.

Figure 5 shows the contour map of sound pressure in the $V_{1}$ free sound field, the total sound field, the conventional recovered sound field, and the $V_{2}$ free sound field on the *S*. The sound pressure distribution of the $V_{1}$ free sound field is different from that of the total sound field. However, the sound pressure distributions of the $V_{1}$ free sound field, the $V_{2}$ free sound field, and the conventional recovered sound field are almost the same.

The mean quadratic errors of the conventional recovered sound field, and the total sound field to the $V_{1}$ free sound field and the $V_{2}$ free sound field are shown in Figure 6. It shows that, at most frequencies, the mean quadratic errors of the total sound field to the $V_{1}$ free sound field and the $V_{2}$ free sound field are over 100%. Meanwhile, the mean quadratic errors of the conventional recovered sound field to the $V_{1}$ free sound field and the $V_{2}$ free sound field are mostly below 10%.

It is clear that the $V_{1}$ free sound field and the $V_{2}$ free sound field are the same in the air because of the weak fluid–structure interaction.

### 3.2. In a Bounded Noisy Underwater Environment

The underwater numerical model is the same as that in the air, but the fluid medium becomes water. The fluid density is 1000 kg/m${}^{3}$, and the sound speed is 1500 m/s. The reference sound pressure and reference sound power are $1\times 10^{-6}$ Pa and $1\times 10^{-18}$ W, respectively. When the cubic shell vibrates in the water, the characteristic quantity λ is greater than 8 in the studied frequency range. Thus, the coupling between the cubic shell and the water is very strong. Considering that the sound pressure radiated by the same vibrating structure in the water is much greater than that radiated in the air with the same vibrational velocity, the strength of the monopole is reduced to −0.002i m${}^{3}$/s at 100 Hz, which is more reasonable. The position of the monopole is unchanged, and the infinite rigid plane can be simply taken as the bottom of the sea.

The power estimator levels directly obtained and recovered are shown in Figure 7. As mentioned above, the power estimator level in the total sound field is very different from in the $V_{1}$ free sound field. The differences are more than 14 dB at most frequencies. The power estimator level in the conventional recovered sound field is significantly different from that in the $V_{1}$ free sound field because the interaction between the vibrating structure and the water cannot be ignored. However, the power estimator levels in the conventional recovered sound field separated by the conventional double-layer method and in the $V_{2}$ free sound field are almost the same. The power estimator level in the new recovered sound field recovered by the proposed method agrees well with that in the $V_{1}$ free sound field.

Figure 8 shows the contour map of sound pressure in the $V_{1}$ free sound field, the total sound field, the conventional recovered sound field, the $V_{2}$ free sound field, and the new recovered sound field on *S*. There are obvious differences between the sound pressure distributions of the $V_{1}$ free sound field, the total sound field, and the conventional recovered sound field. However, the sound pressure distributions of the conventional recovered sound field and the $V_{2}$ free sound field are almost identical. The sound pressure distributions of the new recovered sound field and the $V_{1}$ free sound field share the same distributions.

Figure 9 shows the mean quadratic errors of the conventional recovered sound field, the new recovered sound field, and the total sound field to the $V_{1}$ free sound field and the $V_{2}$ free sound field. The mean quadratic errors of the total sound field to the $V_{1}$ free sound field and the $V_{2}$ free sound field are both over 100%. Influenced by the strong fluid–structure interaction, the conventional recovered sound field to the $V_{1}$ free sound field is also over 100%. However, the mean quadratic errors of the conventional recovered sound field to the $V_{2}$ free sound field are mostly less than 10%. Additionally, the mean quadratic errors of the new recovered sound field to the $V_{1}$ free sound field are mostly less than 40%.

In order to evaluate the robustness of the presented method, the signal-to-noise ratio (SNR) on the *S* is defined as [22]
(26)$$SNR=20log_{10}\frac{\left|\right|P^{f}{\left|\right|}_{2}}{\left|\right|P^{t}{\left|\right|}_{2}}$$
where ||·|| represents the 2-norm of a vector; $P^{f}$ and $P^{t}$ represent the free-field and total sound pressure on *S*, respectively.

The relative error is defined as [22]
(27)$$Error=\frac{\left|\right|P^{rec}-P^{f}{\left|\right|}_{2}}{\left|\right|P^{f}{\left|\right|}_{2}}$$
where $P^{rec}$ represents the sound pressure recovered by the proposed method on the *S*.

The relative errors at ka=0.2, ka=4.0, and ka=8.4 when the strengths of the monopole are −0.0008i m${}^{3}$/s, −0.0032i m${}^{3}$/s, −0.0064i m${}^{3}$/s, −0.0096i m${}^{3}$/s, −0.0128i m${}^{3}$/s, and −0.016i m${}^{3}$/s at 100 Hz are shown in Figure 10. It can be seen from Figure 10 that the relative error decays with increasing SNR and increases with increasing ka. However, the relative error is always less than 11%. This means that the acoustic characteristics radiated by the structure in the free field could be accurately recovered by the proposed method, even for the SNR up to −29.5 dB.

In conclusion, the acquisition of the free-field acoustic characteristics of a vibrating structure is more susceptible to the boundaries and the sources of disturbances in underwater environments. Although the strength of the source of disturbance in the water is nearly ten times smaller than that in the air, the source of disturbance has a more significant influence on the vibration of the structure in underwater environments because the interaction between the vibrating structure and the water cannot be ignored. This means that the $V_{1}$ free sound field and the $V_{2}$ free sound field are quite different. The conventional double-layer BEM can only be used to recover the $V_{2}$ free sound field. However, the proposed method can be used to recover the $V_{1}$ free sound field in a bounded noisy underwater environment.

## 4. Conclusions

The double-layer BEM can only be used to eliminate the incoming sound field and the rigid body scattered sound field. This means that the sound field recovered by this method is the superposition of the free sound field and the elastically radiated sound field. The free-field acoustic characteristics of a structure can be recovered by the double-layer BEM only when the elastically radiated sound field can be ignored such as in the air. However, the elastically radiated sound field cannot be ignored in the water because of the strong fluid–structure interaction. To recover the free-field acoustic characteristics of a vibrating structure in a bounded noisy underwater environment, a method combining the BEM with the vibro-acoustic coupling method was presented. Numerical results show that the acoustic characteristics recovered by the presented method in the bounded noisy underwater environment are the same as those radiated by the underwater vibrating structure in the free field. Meanwhile, the proposed method has good robustness and can be used to accurately recover the free-field acoustic characteristics even for the negative SNR. The realization of this method can break through the limitations of the measurement environment to obtain the free-field acoustic characteristics of underwater vibrating structures. However, it should be pointed out that the presented method only can be used with known structural properties. To remedy this, the method [23] based on pressure measurements for constructing the structural impedance should be introduced into the proposed method. It will be studied to reduce the amount of calculation in the future [31,32,33].

## Figures and Tables

**Figure 1 sensors-21-05521-f001:**
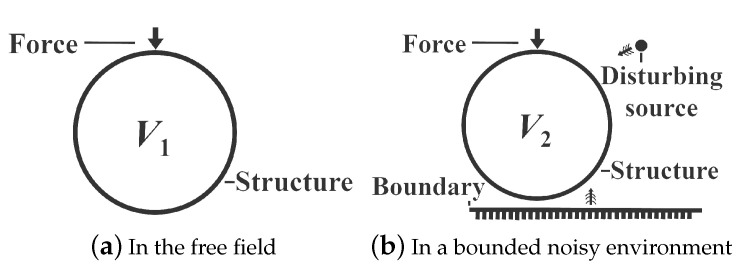
The vibrational behavior of the structure.

**Figure 2 sensors-21-05521-f002:**
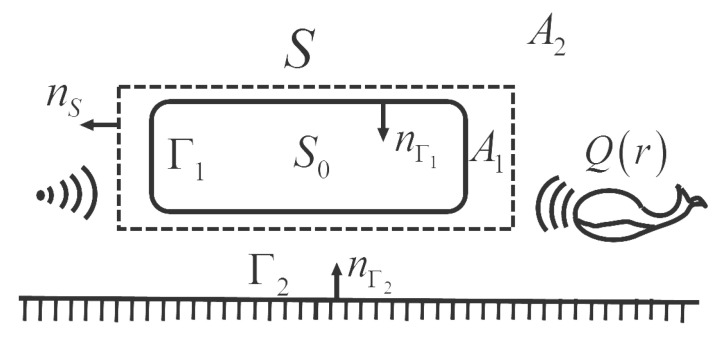
Geometry of a vibrating structure in a complex underwater environment.

**Figure 3 sensors-21-05521-f003:**
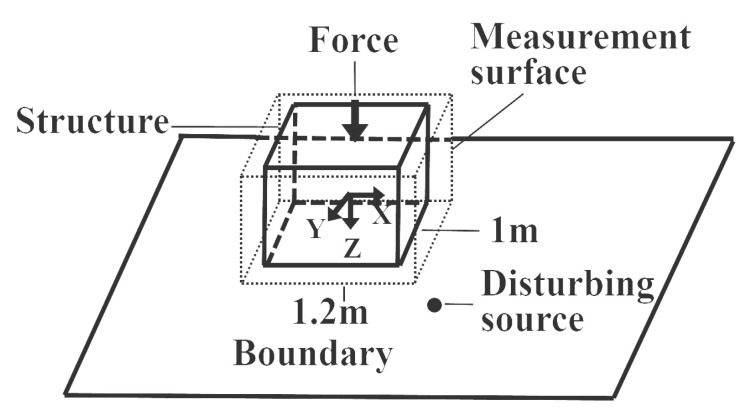
Geometry of the numerical model.

**Figure 4 sensors-21-05521-f004:**
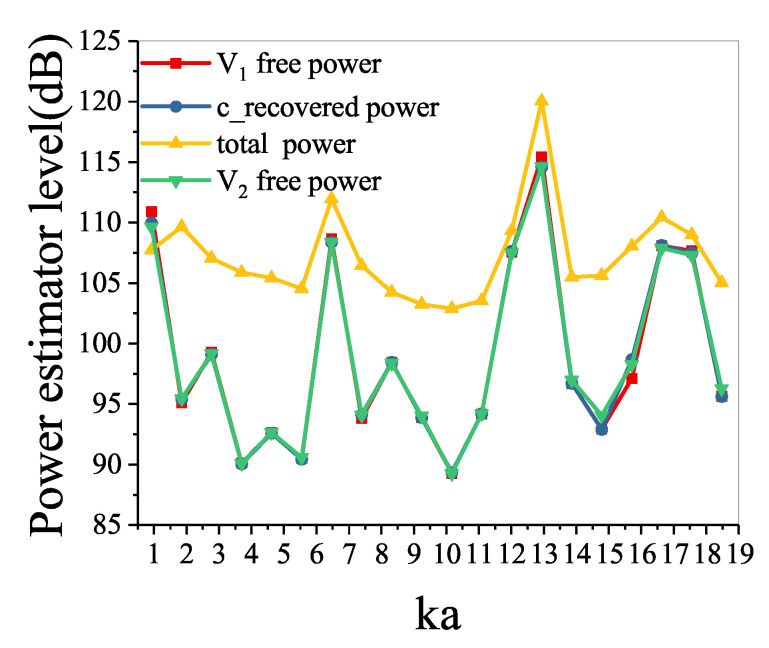
Sound power level on the measurement surface.

**Figure 5 sensors-21-05521-f005:**
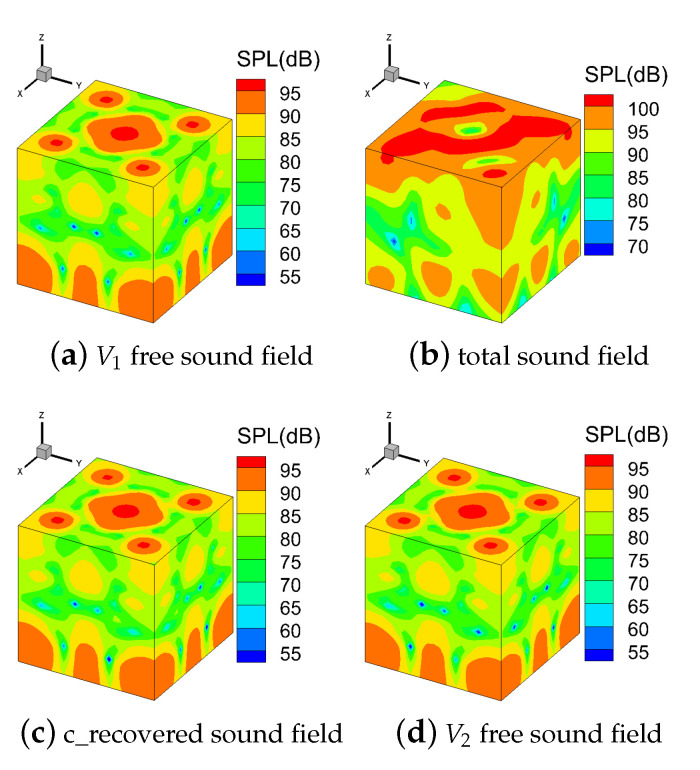
Contour map of sound pressure level (SPL) on the measurement surface at 450 Hz (ka = 8.3).

**Figure 6 sensors-21-05521-f006:**
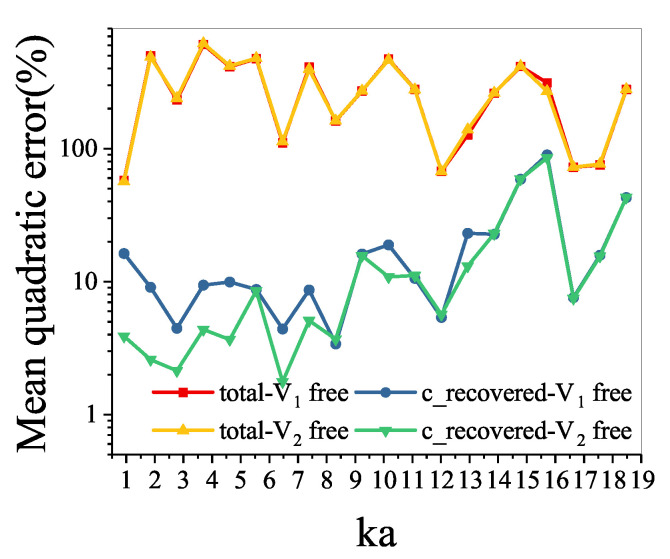
Mean quadratic errors on the measurement surface.

**Figure 7 sensors-21-05521-f007:**
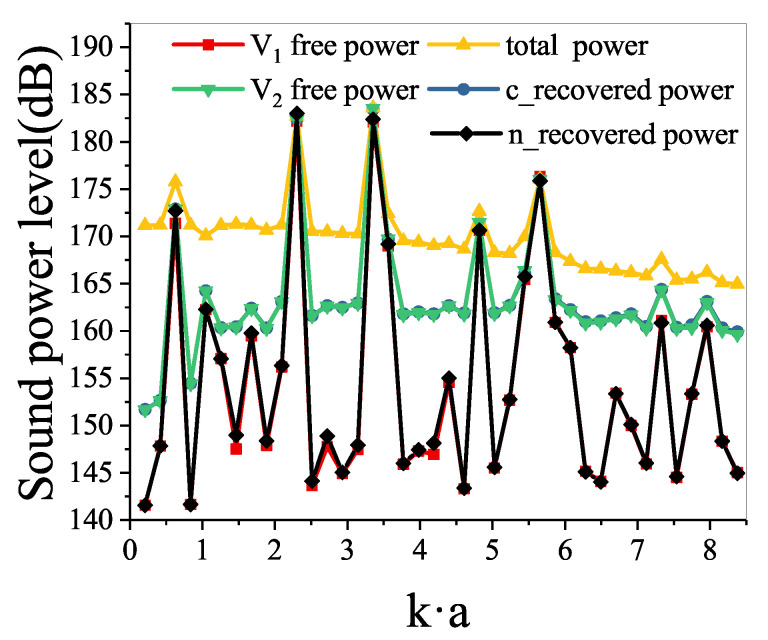
Sound power level on the measurement surface.

**Figure 8 sensors-21-05521-f008:**
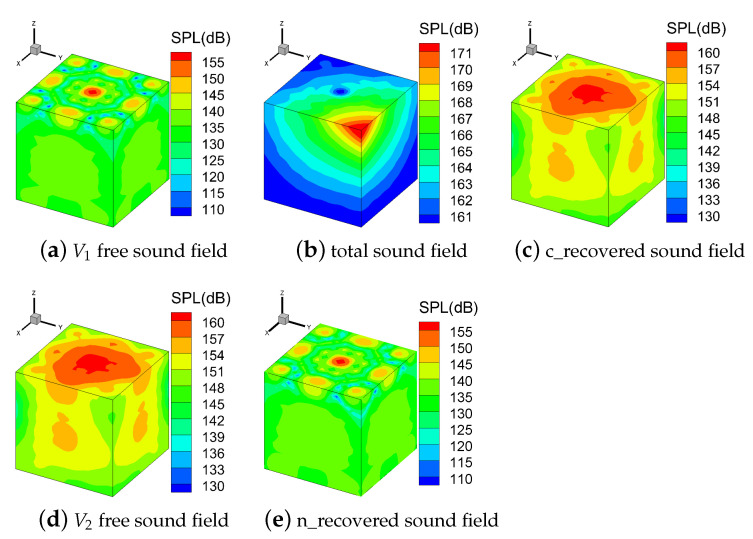
Contour map of sound pressure level (SPL) on the measurement surface at 700 Hz (ka = 2.9).

**Figure 9 sensors-21-05521-f009:**
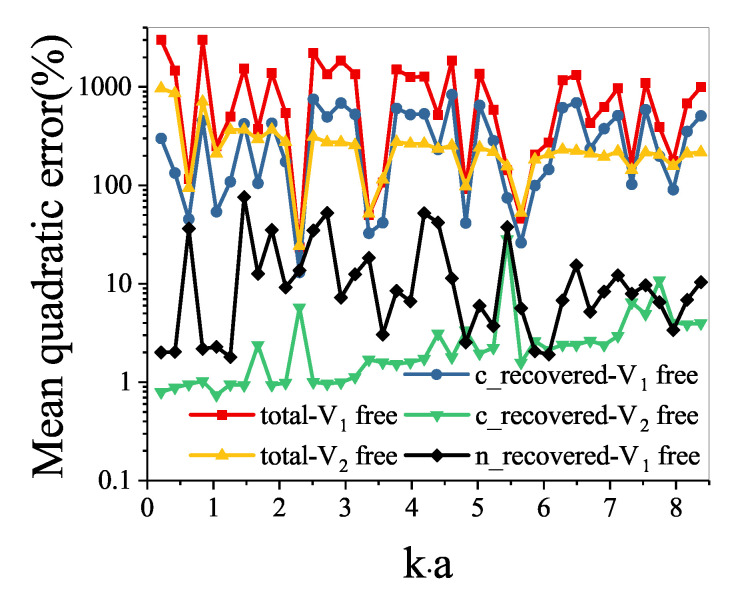
Mean quadratic errors on the measurement surface.

**Figure 10 sensors-21-05521-f010:**
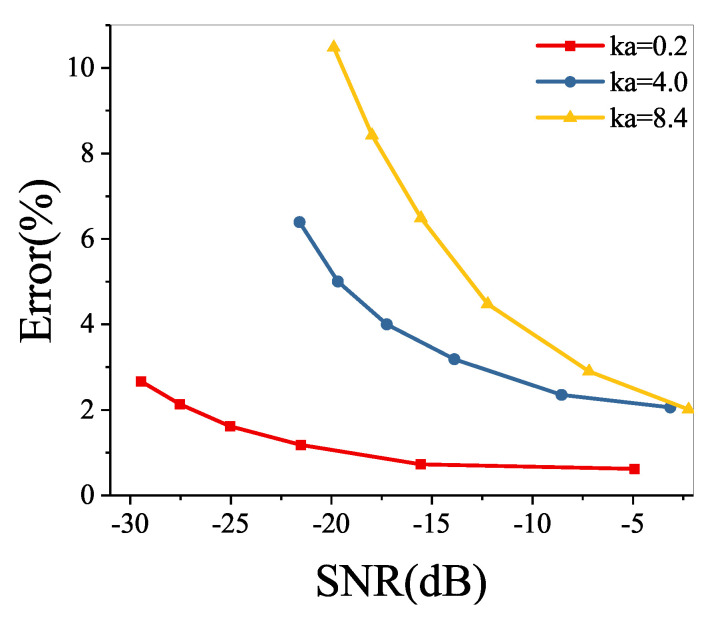
The relative errors of the recovered sound pressure for different frequencies and the SNR.

**Table 1 sensors-21-05521-t001:** The property of shell.

Thickness	Young Modulus	Poisson’s Ratio	Density	Damping
0.005 m	$7\times 10^{10}$ N/m${}^{2}$	0.346	2710 kg/m${}^{3}$	0

## Data Availability

Not applicable.

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
