# Peer review of "Recovering the Free-Field Acoustic Characteristics of a Vibrating Structure from Bounded Noisy Underwater Environments"

_sensors, 2021, doi:10.3390/s21165521_

Round 1
Reviewer 1 Report
My opinion is that it is a very interesting paper. As the authors state, the presented method can recover the acoustic characteristics of a vibrating structure in the free field even when the structure is in a delimited noisy environment.
This method can break the limitations of the measurement environment to obtain the free-field acoustic characteristics of the underwater vibration of a structure (provided the structural properties are known).
Author Response
Thanks the referee for this good advice.
Reviewer 2 Report
The manuscript entitled “Recovering the free-field acoustic characteristics of a vibrating structure from bounded noisy underwater environments" explores the possibility of combining the boundary element method with vibro-acoustic coupling. The paper is generally well written and contributes to the widening knowledge in the field of acoustic modelling. However, the paper requires revisions along the following lines to justify publication.
- How does the presented architecture differ from finite volume-based fluid-structure interaction modelling?
- The literature needs improvement, key studies regarding fluid-structure interaction are now reviewed:
- Arjunan, CJ. Wang, K. Yahiaoui, DJ. Mynors, T. Morgan, VB. Nguyen and M. English (2014). Sound frequency dependent mesh modelling to simulate the acoustic insulation of stud based double-leaf walls. Proceedings of the 2014 Leuven Conference on Noise and Vibration Engineering (ISMA2014). Leuven, Belgium. D/2014/5769/1, ISBN 9789073802919
S.R. Idelsohn, E. Oñate, F. Del Pin, Nestor Calvo. Fluid–structure interaction using the particle finite element method. Computer Methods in Applied Mechanics and Engineering, Volume 195, Issues 17–18, 2006, Pages 2100-2123, ISSN 0045-7825. https://doi.org/10.1016/j.cma.2005.02.026.
- 1 requires further explanation regarding the parameter especially the force field.
- Can the authors present a comparison in discussion showing how this approach compares with other hybrids from literature?
- Did the authors receive any counterintuitive data? Please include this if any. Generally, the influence of critical frequency is an issue.
- What are the limitation and future aspects of this study?
- Did the authors perform any validation?
- The paper is generally well written however further proofreading is recommended to iron out some of the language and grammatical errors.
Reviewer 3 Report
please see comments in the attached file.

Round 2
Reviewer 3 Report
I would just recommend to further check for some refinements to english grammar and possibly further enrich bibliography (e.g., with reference to sentence "The
BEM has advantages in calculating far-field sound pressure and can be used for modeling
the fluid.", by adding a paper with a BEM application highlighting the intrinsic advantages of the methodology for free field analyses, like the one recommended in the following)
- Citarella, M. Landi, Acoustic analysis of an exhaust manifold by Indirect Boundary Element Method, The Open Mechanical Engineering Journal 5 (2011) 138-151."
